# Closed-Loop Uncertainty: The Evaluation and Calibration of Uncertainty for Human–Machine Teams under Data Drift

**DOI:** 10.3390/e25101443

**Published:** 2023-10-12

**Authors:** Zachary Bishof, Jaelle Scheuerman, Chris J. Michael

**Affiliations:** U.S. Naval Research Laboratory, 1005 Balch Boulevard, Stennis Space Center, St. Louis, MS 39529, USA; jaelle.scheuerman@nrlssc.navy.mil (J.S.); chris.michael@nrlssc.navy.mil (C.J.M.)

**Keywords:** uncertainty, interactive machine learning, reinforcement learning, Q-learning, confidence, calibration, human–machine teams

## Abstract

Though an accurate measurement of entropy, or more generally uncertainty, is critical to the success of human–machine teams, the evaluation of the accuracy of such metrics as a probability of machine correctness is often aggregated and not assessed as an iterative control process. The entropy of the decisions made by human–machine teams may not be accurately measured under cold start or at times of data drift unless disagreements between the human and machine are immediately fed back to the classifier iteratively. In this study, we present a stochastic framework by which an uncertainty model may be evaluated iteratively as a probability of machine correctness. We target a novel problem, referred to as the threshold selection problem, which involves a user subjectively selecting the point at which a signal transitions to a low state. This problem is designed to be simple and replicable for human–machine experimentation while exhibiting properties of more complex applications. Finally, we explore the potential of incorporating feedback of machine correctness into a baseline naïve Bayes uncertainty model with a novel reinforcement learning approach. The approach refines a baseline uncertainty model by incorporating machine correctness at every iteration. Experiments are conducted over a large number of realizations to properly evaluate uncertainty at each iteration of the human–machine team. Results show that our novel approach, called closed-loop uncertainty, outperforms the baseline in every case, yielding about 45% improvement on average.

## 1. Introduction

The concept of uncertainty is used with great frequency in machine learning (ML) to provide an understanding of the dependability of model classifications and predictions. However, uncertainty is interpreted and used in many different ways. Uncertainty is used to evaluate the reliability of ML models [1,2], to optimize [3], and to provide transparency to stakeholders [4]. When estimated using entropy, uncertainty is often interpreted as a measure of disorder or randomness [5]. In active learning, various models of uncertainty are used to establish a basis for deciding what examples to query for labeling [6,7]. For applications of interactive machine learning (IML), which involve tightly coupled ongoing interactions between an ML model and a human via a constrained human–computer interface, models of uncertainty may be used to meter the level of work presented to the user [8]. As such, it is often useful to define uncertainty as the probability that a machine’s attempt to solve a problem is incorrect [9,10].

In theory, ML models that have a statistically viable sampling of data may yield accurate values of uncertainty solely based on the distribution of this sampling under a robust data model. However, many problems either do not attain this sampling or suffer from *concept drift,* a modality change in the data context that interrupts or invalidates the data model [11]. This modality change makes it likely that the data model’s yielded uncertainty is low while classification accuracy is also low, which is indicative of an uncertainty model that does not properly reflect the probability of machine correctness. The opposite may also be true, where a yielded high uncertainty is reported during events of high accuracy. This tends to occur when a classifier is not yet adequately trained or when data models are underfit. In the most ideal case, an uncertainty model is automatically calibrated such that its yielded values closely reflect the probability that a misclassification occurs.

A major advantage of the IML paradigm, displayed in Figure 1, is the presence of a human user who may iteratively and immediately correct machine error. In many cases, this feedback may immediately refine a supervised data model by training online on user-corrected information treated as ground truth. Though this technique is trivial for the precision of classification or prediction, very little work has focused on incorporating such feedback to improve the way in which uncertainty is quantified and calibrated [9,12].

The focus of the research presented in this article is to better understand how uncertainty is improved and evaluated for iterative IML applications. A stochastic method for experimentation that takes iterative feedback of machine correctness into account will be presented. To introduce the underlying concepts, a somewhat subjective and generative threshold-selection task is defined and used as a target problem for experimentation. This task involves a human analyst selecting the point at which a decaying signal, namely a sigmoid, is to be considered “low” on a visual plot. As the task progresses, the signals enter new modalities that simulate concept drift. At every step, the machine’s goal is to place the threshold within an accepted tolerance to the analyst’s placement. The machine begins the task at *cold start*, meaning with no prior training data, and trains on the human placement using a supervised model. Additionally, and most importantly for our case, the machine will also provide a probability that this placement is correct. In each iteration of the task, the machine observes the human-placed threshold as the ground truth and trains on the placement for subsequent iterations.

The presented methodology for evaluation examines the machine’s reported uncertainty over many independent realizations of the task in order to compare it to a more accurate measurement of the probability of correctness. Statistics are gathered at every step across all realizations to evaluate the performance of the uncertainty model. We present and compare two supervised models for uncertainty using the presented methodology: a baseline that implements a conventional data-modeled approach using naïve Bayes, and a novel approach using reinforcement learning (RL) to calibrate the baseline using feedback of machine correctness within the reward function. We name the novel approach the *closed-loop uncertainty* (CLU) model because it takes into account machine correctness in an online and iterative manner.

## 2. Related Work

The discussion and formulation of uncertainty is an extensive topic in ML literature. The majority of the prior work discussed in this section focuses on studies that present methods for calculating some value of uncertainty based on the distribution of labeled data. Studies involving uncertainty values that are provided explicitly by humans during training are considered out of the scope of this study.

The interpretation of confidence, which we define to be the opposite of uncertainty, by Pronk et al. [13] for the naïve Bayes classifier formulates the confidence interval for the posterior probabilities of classes. The CLU model we present contrasts this approach in that it takes posterior probabilities as an observable state rather than an estimate for classification confidence.

Defining uncertainty as the probability that a model is incorrect (or confidence as the probability that a model is correct) is useful for evaluating trust in a model or metering the cognitive load and human interaction. However, this definition does have limitations when compared to other discussions in literature. One limitation is that it fails to distinguish between *aleatoric* uncertainty and *epistemic* uncertainty [7,14]. Aleatoric uncertainty involves the distribution of noise and other randomness within the data, while epistemic uncertainty addresses the lack of knowledge within the model. Aleatoric uncertainty is difficult to measure [15], especially under concept drift [16]. Other studies more aligned with our approach have defined uncertainty as a measurement of what is not known at the time of classification [17]. Though this definition allows considerably more leeway, we choose a probabilistic interpretation that allows us to validate models for uncertainty experimentally within an IML paradigm.

Though much of ML theory models classification and prediction on the basis of statistical probability, the general formulation and evaluation of uncertainty is considered something of an afterthought [17]. The general idea behind most models of uncertainty is a mathematical basis for the data model. For example, the softmax layer in a neural network produces a score that may indicate uncertainty about a classification in multiclass problems [2]. However, many times such models have demonstrated some sort of best fit to the probabilities of a label in training data rather than a logical measurement of machine knowledge about a specific class [18]. Additionally, these models are often not well calibrated or may not adequately reflect the probability of the machine to be incorrect after calibration [19]. In these cases, a classification with a low uncertainty does not necessarily imply a high accuracy [20]. For streaming problems, these shortcomings of conventional ML models for uncertainty are exacerbated in the presence of concept drift, where data modalities in the stream may change abruptly or unexpectedly [16]. Therefore, it may be worthwhile to consider if an interpretation of uncertainty that departs from stringent mathematical definitions for the data model could be practical in providing a more accurate quantification of uncertainty, especially in situations where sampling is low or concept drift is expected to occur.

Entropy as a means of estimating uncertainty has been explored, for example, as an effective method for variable selection in naïve Bayes classification [21], as a component of the reward function in building robust RL [22], and in the context of neural networks for image processing [15]. Some of these techniques use variance and entropy of a statistical distribution to measure uncertainty, which could also aid in detecting concept drift [23]. Our CLU model differs from these techniques in that it does not explicitly detect concept drift or variance in input feature data. Instead, CLU observes some measurement of uncertainty, namely the posterior probabilities of a black-box classifier, and calibrates based on the observed accuracy of the machine via the feedback from the analyst. Therefore, the goal of our work is not to detect concept drift, but to provide a model of uncertainty that is adaptable to concept drift when the drift causes bias in the underlying model.

The CLU model presented in this study employs a feedback model to achieve an improved quantification of uncertainty, which may itself have a higher order of uncertainty associated with it. Such phenomena for reinforcement learning have been discussed thoroughly in Clements et al. [24], where the difference between aleatoric and epistemic uncertainty is distinguished within deep reinforcement learning. The study builds on previous work that aims to view uncertainty through the lens of a return distribution and the variance associated with it [25,26]. These previous studies in reinforcement learning have defined uncertainty as a function of the input data or lack of input data, while the CLU implementation that we present defines uncertainty as a function of the accuracy of the underlying classification model within a stochastic process where accuracies may be measured.

IML models have calculated uncertainty using mixture models or some approach that resembles that of the machine learning algorithm used in the classification process [27]. In other studies, uncertainty has been defined as the sample conditional error, which is the probability that the classifier makes a mistake on a given sample [28,29]. These techniques require that the underlying distributions of the data model are known, which is often not possible. Our approach allows for a black-box baseline model of uncertainty, and we show that it performs accurately even when the data-model distributions are not accurate.

## 3. Background

A sigmoid curve, or logistic curve, is an S-shaped curve. For our setting, we are interested in a version that begins near y=1 when x=0 and decays to approach y=0 as *x* approaches 1. The sigmoid is centered somewhere between x=0.2 and x=0.8. In general, we can describe this curve using the following function:(1)$$y\left(x\right)=\frac{1}{1+e^{k(x-x_{0})}}$$
where $x_{0}$ is the curve’s midpoint and *k* is the decay rate.

Gaussian naïve Bayes classification is used as a baseline, which operates by calculating posterior probabilities assuming all predictor covariates have a normal distribution and are independent [30]. However, it is possible to construct an effective naïve Bayes classifier when the independence assumption is alleviated [31,32]. The feature spaces used in the naïve Bayes classifiers discussed in this paper do not contain independent features.

Kernel density estimation (KDE) can be used to estimate the distribution of a covariate, which is useful in addressing problems posed by continuous variables. Namely, KDE allows for smoothing the distributions of the feature space [33].

A Markov decision process (MDP) is defined as a tuple (S,A,ξ,R,γ), which comprises a state space *S*, a set of possible actions *A*, a transition probability matrix ξ, a reward function *R*, and a discounting factor γ. [34]

Q-learning is a type of model-free RL used to find the optimal policy, which is the strategy for choosing actions in each state, by updating the expected reward for state–action pairs. The only conditions for convergence on the optimal policy is that the actions are repeatedly sampled in all states and the action-values are discrete [35]. This allows convergence towards the optimal policy by choosing actions randomly for each trial. Q-learning is a popular implementation of RL due to its simplicity and effectiveness at finding optimal solutions in MDPs.

## 4. Materials and Methods

All experimentation and machine models were implemented and run in the R programming language. Naïve Bayes classification and posterior probability calculation were performed using the ‘naivebayes’ package [36], and all reinforcement learning was conducted using the ‘ReinforcementLearning’ package [37]. The code used is available upon request from the corresponding author.

### 4.1. Threshold Selection Problem

The methodology for uncertainty presented in this study is specific to an online IML implementation [38]. Though a multitude of problems for which IML implementations exist are available in the current state of the art, the problems either possess overly complex features and interfaces or do not provide controls to induce concept drift or realize parallel tasks in a stochastic manner. We present a threshold selection problem that exhibits the properties of being an intuitive task definition with a simple interface, a 2-dimensional dataset with a minimal feature space, a stochastic basis for generating a very large number of 2D examples for the problem space, moderate subjectivity that prevents trivial solutions without human interaction, and a parameterized complexity and noise model used to induce concept drift.

This threshold selection problem is a simpler surrogate for online IML applications in the sense that it exhibits subjectivity in preference and concept drift, similar to those discussed by Kabra et al. [39] and Michael et al. [10]. Additionally, it is straightforward to define this problem as a stochastic process by which uncertainty may be studied and evaluated as a probability of correctness. The problem is able to be realized such that a single trial of the problem will, for the most part, have similar complexity across all realizations.

#### 4.1.1. Trial Definition

A *trial* consists of a 2D plot of a decaying sigmoid curve sampled at 100 regular discrete points. The user is asked to locate the point at which they think the sigmoid transitions from a high state to a low state, i.e., the *human-placed threshold*, which is treated as ground truth. Increasingly complex noise is added as the trials go on to induce concept drift. Examples of these trials in each phase of the experiment are displayed in Figure 2.

The user is constant and maintains their preference within a given realization of the experiment. Other realizations may contain different preferences for the threshold placement. The location where the signal is no longer considered to be in the “high” state is a subjective choice made by the analyst (see Section A.5).

#### 4.1.2. Phase Definition

In a realization of the problem, multiple trials are generated and presented to the human–machine team in a particular order. A *phase* is a consecutive subset of trials with similar stochastic parameterization, which defines a fixed modality of complexity. Overall, the progression of phases are designed with the intent of generating more complex sigmoids.

Phase I contains only square wave trials, making it the simplest phase with minimal subjectivity. The only stochastic parameter is the center of the plot, $x_{0}\in \left[0.2,0.8\right]$, which is selected in a uniform random way. The curve depicted for the analyst in each trial for Phase I is
(2)$$y\left(x\right)=\left\{\begin{array}{cc} 1 & \;\mathrm{if}\;x\leq x_{0}\\ 0 & \;\mathrm{if}\;x>x_{0} \end{array}\right.$$

Phase II depicts a logistic curve, as defined in Equation (Equation 1), where the decay rate *k* is randomly assigned. This causes the curve to lead to a slower or faster decay:(3)$$k={\left(1-\frac{1}{1+\exp(\frac{-1}{1-d})}\right)}^{-1}$$
where d∈[0,1) is a uniform random variable. This formula was found to yield the best mix of sigmoid decay after experimenting with a variety of other approaches.

The subsequent phases, III-V, induce random noise into the sigmoid. The noise *N* is applied in an additive way. The magnitude of the noise is N=mcos(τπ), where τ∈[0,1] and m∈[0,0.8] are uniform random variables. The variable *m* indicates the maximum magnitude of additive noise such that N∈[−m,m]. Trials of Phase III generate sigmoids with *N* added to a uniformly random percentage of uniformly random chosen points. Phase IV generates sigmoids with *N* added to randomly chosen sub-intervals of points. Sub-interval sizes are chosen by the uniform random variable l∈[0,100]. The number of sub-intervals is randomly chosen from the set $\left\{n\in I|0\leq n\leq \left\lfloor \frac{100}{l}\right\rfloor \right\}$. Finally, Phase V sigmoids contain additive noise, *N*, throughout the entire function. Depending on the parameters and noise, any phase can potentially generate a plot resembling that of a previous phase.

Figure 2 illustrates an example trial from each phase. The human-placed threshold, which is considered ground truth, is shown in blue, and the machine-placed threshold, which is placed using a naïve Bayes classifier, is shown in red. As will be discussed in Section 4.2, we examine different tolerances for machine-placed thresholds to be considered correct.

### 4.2. Na ïve Bayes Classifier

A Gaussian naïve Bayes classifier is used to predict the location of the human-placed threshold. We refer to this predicted location as the *machine-placed threshold*, which is shown as the red line of the example plots in Figure 2. The features used by the naïve Bayes classifier include the basic coordinates of each trial’s sigmoid. In addition, several features are extracted to enhance the feature space. This includes the first derivative of the sigmoid at each discrete point, the second derivative of the sigmoid at each point, the mean of the next 10 points’ y values, and the mean of each point along with its 2 direct neighbors. These features were determined using conventional feature selection techniques, and they were shown to yield generally high accuracy in each phase after warm-up for reasonable tolerances of machine placement.

Labels are determined by assigning 1 to all sample points with x-coordinates less than or equal to the human placed threshold and 0 to all sample points with x-coordinates greater than the human placed threshold. This labeling scheme worked best for machine placement when compared to other schemes, mainly due to its relatively balanced positive and negative labels when training.

The location of the machine placement was chosen at the first point with the mean of the posterior probability (generated from Gaussian naïve Bayes) and that of its two direct neighbors being greater than 0.55.

When analyzing the performance of naïve Bayes, it was found that the retention of all training instances hurt performance in later phases due to bias. The most optimal classifier only considered the instances of the 6 preceding trials when training, which is discussed in greater detail in Section A.2.1. This type of forgetting is typically known to be beneficial for drifting data streams [40], and thus was used for our experimentation.

### 4.3. Baseline Uncertainty Model

A naïve Bayes classifier is used as a *baseline uncertainty model* to produce first-order input uncertainty values for CLU. We use a baseline in this manner because most supervised ML implementations naturally choose to formulate uncertainty directly from the data model, especially in the generative case.

The input to the CLU model we present may take any black-box baseline uncertainty model. The Gaussian naïve Bayes model we use in this study yielded the best accuracy for uncertainty as a probability of correctness over various other relatively simple models we have observed. For an exploration of these, see Section A.4.

Unlike the model for machine placement, the uncertainty model must account for some *placement tolerance*, denoted by the variable *T*, allowed by the application. The placement tolerance is an independent variable that determines the distance within which a machine placement must be from a human placement to be considered correct. The lower the placement tolerance, the lower the expected placement accuracy, and vice versa.

In labeling a trial for the uncertainty model, all sample points within the distance of the placement tolerance are labeled as 1, and all other points as 0. Using this model, the posterior probability that a sample point in a trial should be given a label of 1 may be calculated in the conventional way using the product of prior probability and likelihood. A confidence can then be generated by taking the average of all these posterior probabilities within the placement tolerance of the machine-placed threshold.

Other than labeling, the baseline uncertainty model differs from the naïve Bayes classifier in two other ways. The first is the absence of instance forgetting, since it was found in experimentation that forgetting does not improve the accuracy of the uncertainty model for the threshold selection problem. Second, the first- and second-derivative features are removed from the feature space for the uncertainty model. These features generate a large amount of bias for the way in which thresholds for correctness are labeled, as was found through feature selection.

### 4.4. Closed-Loop Uncertainty

The CLU model we present employs reinforcement learning to observe the baseline confidence value and machine correctness at each iteration in order to refine the baseline model. This refinement is made possible through the incorporation of correctness (via human intervention) into the reward function.

The structure of CLU is displayed in Figure 3. The baseline uncertainty model generates a state space with modifications made to the baseline uncertainty values based on the action space. Rewards are generated using the ground truth provided by human intervention to estimate the calibration of the baseline uncertainty model. These components constitute the core elements of the MDP, and a policy for calibration is constructed using Q-learning.

In the following sections, the Markov decision process and the policy selection will be defined in detail. The size of the state space, action space, and the window parameter in the reward function were selected using a sensitivity analysis described in Section A.2.

#### 4.4.1. State Space

Define the state space *S* by taking a discretization of the difference between successive confidence values
(4)$$\Delta C_{t}=C_{t}-C_{t-1}$$
where *t* indicates the trial number within a realization. Each $C_{t}$ is a confidence value, so $\forall t,C_{t}\in \left[0,1\right]$ and thus $\Delta C_{t}\in \left[-1,1\right]$. $S_{t}\left(\Delta C_{t}\right)$ maps the interval [−1,1] onto the set {1,2,3}, and can be expressed as
(5)$$S_{t}\left(\Delta C_{t}\right)=\left\lfloor \frac{3\Delta C_{t}}{2}+\frac{5}{2}\right\rfloor$$

This process of discretization is employed to ensure convergence on the optimal policy, which is described in greater detail in Section A.3. Conventional discretization techniques resemble this approach [41].

#### 4.4.2. Action Space

The set of possible actions
(6)A={x∈Q|−1≤x≤1,x=b/7,b∈I}
is defined by evenly discretizing the interval [−1,1] into 15 values.

Given an action $A_{t}$ and a confidence value $C_{t}$, we can find the confidence value that is the result of an action:(7)$$C_{t}^{\prime }\left(A_{t},C_{t}\right)=\left\{\begin{array}{cc} C_{t}+A_{t}C_{t} & \mathrm{if}\;A\leq 0\\ C_{t}+A_{t}\left(1-C_{t}\right) & \mathrm{if}\;A>0 \end{array}\right.$$

This definition allows for a fractional shift in any given state’s confidence value either up or down. For example, an action of $\frac{1}{7}$ can be thought as the act of making a $\frac{1}{7}$ gain of any given state’s confidence value towards 1, while the action $-\frac{1}{7}$ would cause a $\frac{1}{7}$ loss towards zero.

#### 4.4.3. State Transitions

Let $S_{t}^{\prime }$ be the state reached from $S_{t}$ as a result of action $A_{t}$, and
(8)$$\Delta C_{t}^{\prime }=C_{t}^{\prime }-C_{t-1}$$

The state $S_{t}^{\prime }$ is calculated using Equation (Equation 5), and thus becomes
(9)$$S_{t}^{\prime }=S\left(\Delta C_{t}^{\prime }\right)$$

The transition probabilities ξ are not needed ahead of time due to the usage of model-free reinforcement learning [34].

#### 4.4.4. Reward Function

For CLU, an ideal reward function would give a higher reward when the policy is choosing actions that tune the bias of the confidence in the direction of the probability of a correct classification. For this reason, the reward function uses an estimate of the error between the confidence of the baseline model and that of the accuracy of the machine placement for a fixed window size.

Define accuracy $p_{t}$ at trial *t* to be equal to either 1 when the machine placement is correct or 0 when the machine placement is incorrect. The mean streaming accuracy, which measures the fraction of machine placements that are deemed correct within a given window $W_{t}$, where
(10)$$W_{t}=\left\{j\in N|\max\left(1,t-w-1\right)\leq j\leq t-1\right\}$$
is defined as
(11)$$\bar{p_{t}}=\min{\left(w,t\right)}^{-1}\sum_{i=\max(1,t-w-1)}^{t-1}p_{i}$$
where the window size *w* is set equal to 24 (for analysis of the window size parameter, see Section A.2.2).

Given a present policy $\pi_{t}:S\to A$ at trial *t*, a state $S_{t}$, and a window size *w*, the window mean squared error, which estimates the discrepancy between machine placement accuracy and baseline confidence values, is defined as
(12)$$D_{t}^{\prime }=\min{\left(w,t-1\right)}^{-1}\sum_{i=\max(1,t-w-1)}^{t-1}{\left(C_{i}^{\prime }\left(\pi_{i}\left(S_{i}\right),C_{i}\right)-\bar{p_{t}}\right)}^{2}$$

For the baseline confidences, i.e., the policy π(s)=0 for all *s* and $C_{t}^{\prime }=C_{t}$ at every *t*, the baseline window mean squared error $D_{t}$ is
(13)$$D_{t}=\min{\left(w,t\right)}^{-1}\sum_{i=\max(1,t-w-1)}^{t-1}{\left(C_{i}-\bar{p_{t}}\right)}^{2}$$

These are used to calculate reward $R_{t}$ for transitioning from state $S_{t}$ to $S_{t}^{\prime }$ due to action $A_{t}$.
(14)$$R_{t}=\left\{\begin{array}{cc} 1-D_{t}^{\prime } & D_{t}^{\prime }<D_{t}\\ -D_{t}^{\prime } & D_{t}^{\prime }\geq D_{t} \end{array}\right.$$

Note that at each trial *t*, $D_{t}^{\prime }$ depends on not only the mean streaming precision $\bar{p_{t}}$, but the present policy at each trial. This present policy is used to define $C_{t}^{\prime }$ for all trials in a given window $W_{t}$. The values for $C_{t}^{\prime }$ inside each $W_{t}$ depend on the states and baseline confidences for each trial inside $W_{t}$, yielding a unique $R_{t}$.

#### 4.4.5. Policy Selection

At each trial, all of the previous trials’ states and black-box output confidences are stored, and actions are randomly assigned to each trial. This forms a basis for the online Q-learning model. The action-value function Q(S,A) gives the expected reward for each state under each action.

The learning rule for Q-learning is defined at each trial *t* as
(15)$$\begin{array}{cc} Q(S_{t},A_{t}) & \leftarrow Q\left(S_{t},A_{t}\right)+\lambda \left(R_{t}+\gamma \max_{a}Q\left(S_{t}^{\prime },a\right)-Q\left(S_{t},A_{t}\right)\right) \end{array}$$
where γ,λ∈[0,1] are the discount factor and learning rate, respectively [34]. Both parameters are set to 0.1.

Given a state *s*, the optimal policy $\pi_{*}$ yields the action that produces the highest estimated Q-value. If all Q-value estimates are either negative or zero, then the optimal policy is to take action “0”.
(16)$$\pi_{*}\left(s\right)=\left\{\begin{array}{cc} \arg\max_{a}Q\left(s,a\right) & \exists a\in A\;s.t.\;Q(s,a)>0\\ 0 & \mathrm{otherwise} \end{array}\right.$$

Given a state $S_{t}$ and randomly chosen action $A_{t}$ at trial *t*, the present policy, $\pi_{t}$, replaces the optimal action, $\pi_{*}\left(S_{t}\right)$, with $A_{t}$:(17)$$\pi_{t}\left(s\right)=\left\{\begin{array}{cc} A_{t} & s=S_{t}\\ \pi_{*}\left(s\right) & s\neq S_{t} \end{array}\right.$$

The present policy is used to determine reward values, as described in Equation (Equation 12). The optimal policy at each trial is used to calculate the confidence values as shown in Figure 3.

### 4.5. Methodology of Evaluation

In order to evaluate confidence as a probability of correctness, the experimentation must be conducted as a stochastic process. Judging an uncertainty model based on a single realization of an experiment is likened to evaluating the fairness of a coin based on a single flip. Therefore, the presented experimental methodology for evaluating uncertainty is based on many parallel realizations of the experiment that are evaluated at every trial *t*. Preferences from user to user are not expected to be similar, but each user’s preference is expected to be consistent within a realization.

Results are reported by subtracting the mean absolute error (MAE) from 1 at each trial *t* across all realizations:(18)$$1-MAE\left(t\right)=1-\frac{1}{n(C_{t})}\sum_{c\in C_{t}}\left|P_{t}-c\right|$$
where $C_{t}$ is a set containing confidences at trial *t* across all realizations, $n(C_{t})$ represents the number of realizations in $C_{t}$, and $P_{t}$ is the accuracy of machine placement at trial *t* across all realizations. Subtracting from 1 simplifies interpretation, with higher values of 1−MAE indicating better model performance.

Experiments were conducted for 30 realizations, each consisting of the 5 phases described in Section 4.1.2. Each phase contains 7 trials, totaling 35 trials per realization. This number of realizations was found to reach a statistically significant sample size during experimentation. Ground truth was labeled for each trial of each realization by a user who was asked to maintain preference for the entire realization. All machine models began each realization with a cold start, meaning that all classifiers and uncertainty models began the first trial of each realization with no training data. As discussed in Section 4.2, the classifier used for machine placement disregards all training prior to the 6 most recent trials, as this improved the accuracy of machine placement. The baseline uncertainty model did not implement this type of forgetting since forgetting did not yield improved results for uncertainty.

## 5. Results

Figure 4 shows the average results across all realizations for tolerances 0.02, 0.1, and 0.2 individually. The plots include the machine placement accuracy for reference. These particular plots are chosen as they compare and contrast the models for situations where machine placement accuracy is expected to be low (T=0.02), moderate (T=0.1), and high (T=0.2). Figure 5 shows results for each trial averaged across all tested tolerances, which include all values in the set ranging from 0.02 to 0.20 in increments of 0.02. T=0 was excluded because it requires the machine placement to be identical to that of the human placement, which, in a continuous space, is impractical. Tolerances greater than 0.2 cover nearly half of a trial graph or more, which becomes less meaningful when evaluating uncertainty.

The CLU model significantly improves upon the baseline in almost every trial; however, the most noticeable trials where the baseline performs better than CLU are the first trials of phases II and III. There are two important reasons why this behavior is observed. First, the baseline model for uncertainty tends to be biased towards underconfidence, meaning that its reported probability of correctness tends to be much less than the machine placement accuracy in general. This is also true for higher values of tolerance, where classifier accuracy is expected to be high. Because Phases II and III induce a relatively high amount of concept drift, the machine placement accuracy suffers greatly in the first trial of these phases. As the baseline is biased towards underconfidence, any significant fall in accuracy will cause it to exhibit a higher 1−MAE. The second reason that the baseline model outperforms CLU in these particular trials is CLU is unable to detect concept drift from first-class features of the data. As it is driven by feedback of correctness, it must observe that its current policy is inaccurate by observing that the incorrect machine placement was corrected by the human after entering a new phase of drift. However, as the machine placement accuracy improves incrementally in the subsequent trials of Phases II and III, the CLU model is able to very quickly outperform the baseline by accounting for the baseline’s underconfidence by taking into account the incorrect machine placements. Phases IV and V do not generally induce as much concept drift, as indicated by the machine placement accuracy of those trials. Additionally, 1−MAE for CLU in Phase I, which is the simplest phase where machine placement is 100% accurate, is lowest among all phases. This is mainly due to the lack of training information from cold start as well as the fact that the bias of the baseline continues to decrease in this phase, undoing the adjustment of the CLU algorithm.

The baseline model (the red line) dramatically under performs during phase I when the machine placement accuracy is at 100 percent, due to the features acting discrete for this phase, while Gaussian naïve Bayes performs best for continuous normally distributed features.

A summary of 1−MAE statistics for various tolerances are shown in Table 1. These values were calculated across all trials of all realizations of experiments. As shown, CLU is able to improve upon the baseline model by up to 68% and exhibited a 45% average improvement overall. In general, better CLU performance trended towards tolerance values of 0.06–0.16, where the machine placement accuracy was expected to be moderate, and the baseline performed most poorly.

## 6. Discussion

By considering feedback on machine correctness within an iterative IML paradigm, it has been shown that uncertainty modeling can greatly improve upon a baseline that only considers the input data model. A novel closed-loop uncertainty model exemplifying this improvement was presented. This implementation uses reinforcement learning to incorporate feedback on machine correctness and calibrate a more traditional uncertainty model. A simple yet effective threshold selection problem, which exhibits many of the important characteristics of IML such as concept drift and subjectivity, was presented and used for experimentation.

The calibration of uncertainty under concept drift can be performed effectively by incorporating human feedback into the reward function of a reinforcement learning model. However, the performance of this calibration is dependent on how the states, actions, and rewards are defined. If the state space does not reflect the baseline uncertainty model, if the action space is too complex, or if the reward function does not incentivize actions in the direction of better calibration, it will result in slowly or poorly calibrated uncertainty values. Therefore, careful consideration is required when defining these components of the MDP, and finding the most optimal definitions may not be straightforward. It is possible that the MDP presented for the threshold selection problem could be enhanced by designing a more nuanced reward function.

The work presented in this paper is meant to demonstrate the concept of modeling and evaluating uncertainty for IML problems. Though the threshold selection problem is interesting and useful for this type of demonstration, the problem itself is relatively simple in terms of classification and feature space. The approach of calibrating uncertainty using the architecture of CLU is presented in the context of a two-class problem. However, there is no reason why CLU cannot generalize to a multi-class problem provided the multi-class problem has human feedback indicating ground truth and an initial baseline uncertainty, which could be set equal to a constant value as discussed in Section A.4.3. The reward function would need to be altered to account for the effect changing the uncertainty of one class would have on another class. A more complex problem would most likely necessitate a more complex feature space and thus a finer-grained state–action space requiring higher-level optimization. In such cases, algorithms like DQN could be effective [42]. Our group is currently working to extend this research to the challenging domain of geographic region digitization. In these more complex problems, the cognitive load of the human analyst must be taken into consideration. Too little cognitive load, and the machine model is slowing the analyst down; too much, and the analyst becomes overwhelmed and will most likely abandon the machine teammate. As such, we are currently exploring more complex methodologies that analyze uncertainty modeling as a control process with the threshold of failure determined at every iteration.

The detection of concept drift, as demonstrated in Du et al. [23], could greatly improve the precision of uncertainty models for iterative streaming problems. We are currently working towards incorporating active learning methods for detecting drift into the closed-loop uncertainty model.

## 7. Patents

A provisional patent has been filed for the system and method of uncertainty feedback for an iterative interactive machine learning model.

## Figures and Tables

**Figure 1 entropy-25-01443-f001:**
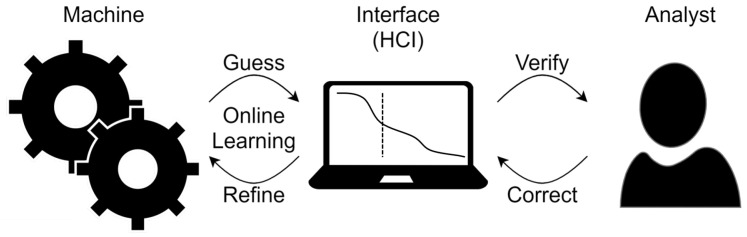
The interactive machine learning paradigm tightly couples machine learning to a human analyst via an intuitive interface for a task.

**Figure 2 entropy-25-01443-f002:**
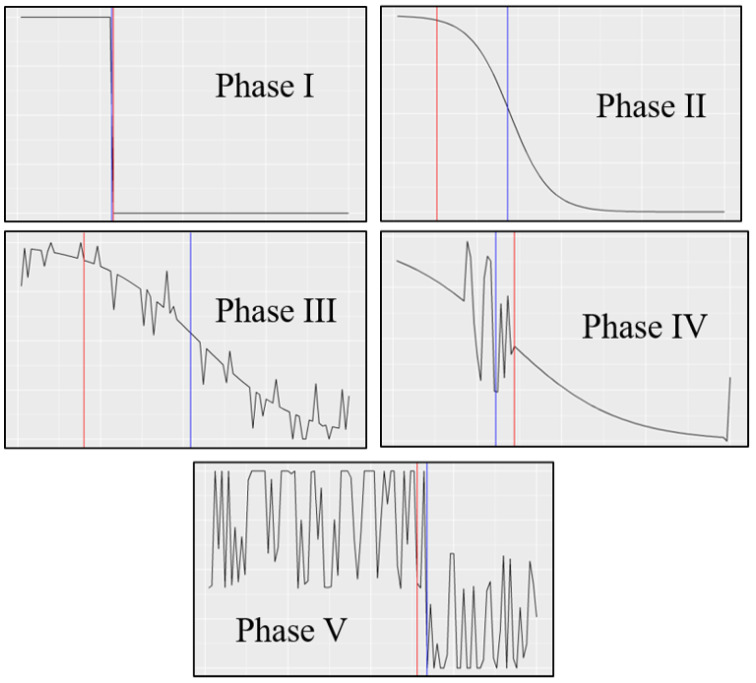
An example trial of each phase showing the human-placed threshold in blue and the machine-placed threshold in red. Correct machine placement is determined by the tolerance, where a tolerance of 0.04 would mark the phases I and V examples correct and all others incorrect. Human-placed thresholds are subjective due to the preference of the analyst.

**Figure 3 entropy-25-01443-f003:**
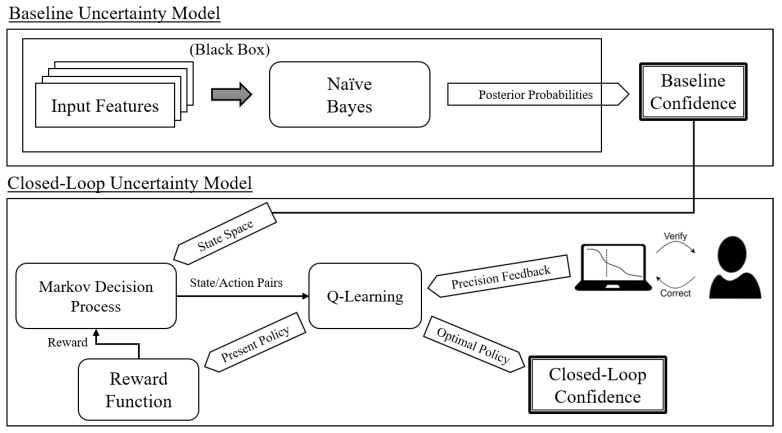
Architecture of closed-loop uncertainty.

**Figure 4 entropy-25-01443-f004:**
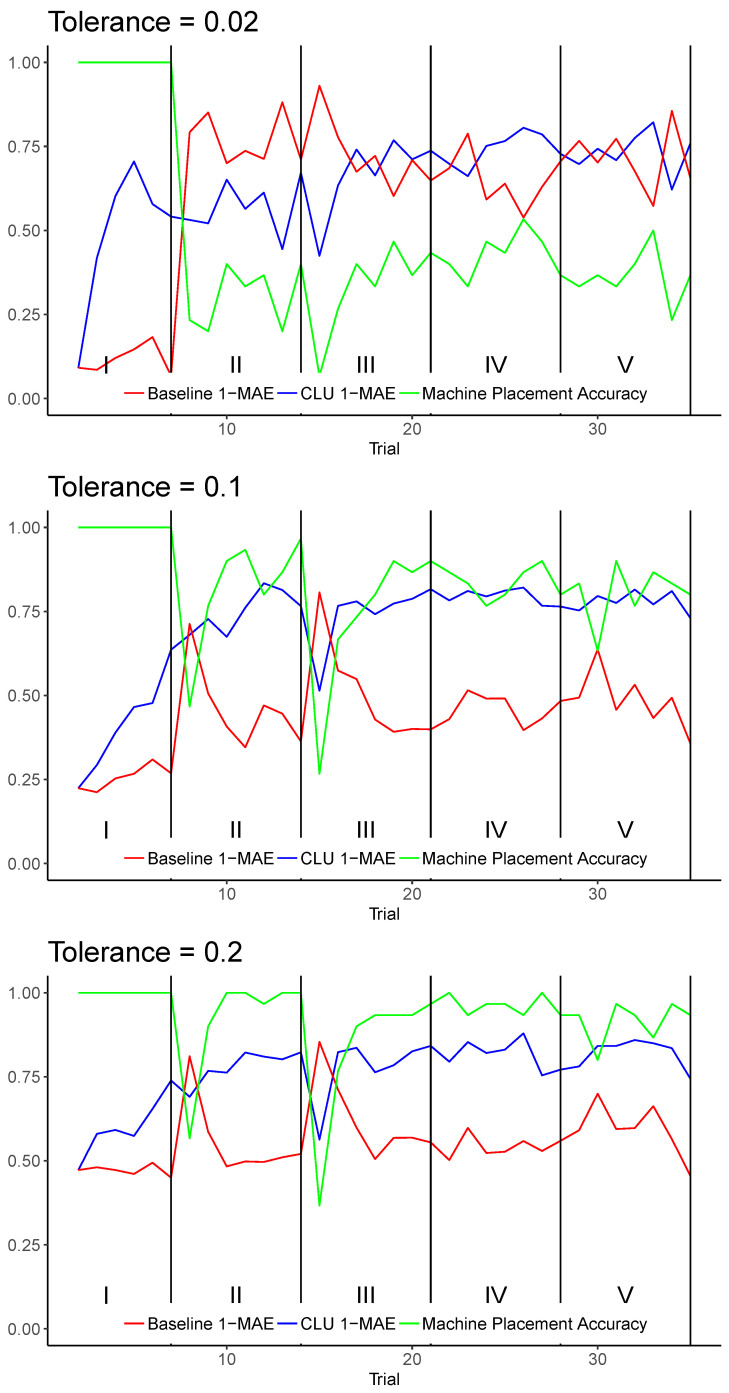
Results comparing the baseline uncertainty model to the CLU model for tolerances 0.02, 0.1, and 0.2, respectively, from top to bottom. Results are plotted for each trial across all realizations. The green plot shows the average accuracy of machine placement.

**Figure 5 entropy-25-01443-f005:**
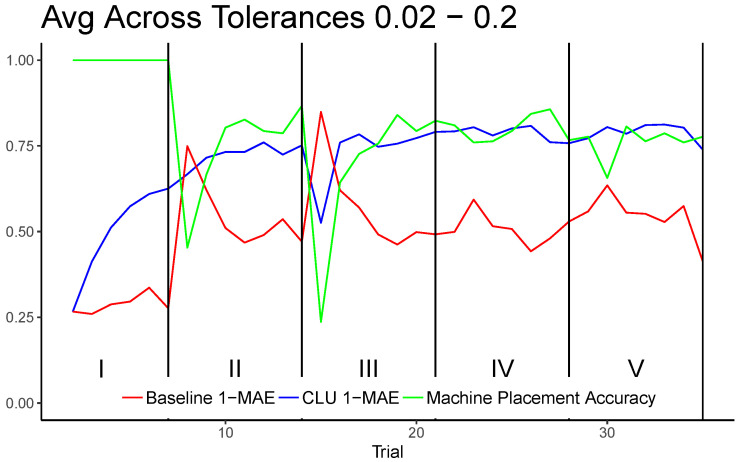
Results as shown in Figure 4, but averaged across all explored tolerances.

**Table 1 entropy-25-01443-t001:** Results of 1−MAE for CLU and baseline models. The mean and variance across all trials and realizations for several tolerances are shown, as well as the accuracy of the machine placement.

		1−MAE			
		**CLU**		**Baseline**	
**Tolerance**	**Accuracy**	**Mean**	**Variance**	**Mean**	**Variance**
0.02	0.471	0.645	0.0213	0.609	0.0614
0.04	0.625	0.675	0.0231	0.507	0.0382
0.06	0.734	0.676	0.0185	0.444	0.0241
0.08	0.796	0.718	0.0203	0.427	0.0202
0.1	0.832	0.704	0.025	0.44	0.0167
0.12	0.857	0.729	0.0176	0.463	0.0154
0.14	0.879	0.742	0.0163	0.485	0.0127
0.16	0.898	0.753	0.0141	0.509	0.0116
0.18	0.908	0.727	0.0104	0.537	0.0105
0.2	0.923	0.764	0.0102	0.56	0.00898

## Data Availability

The data presented in this study are available on request from the corresponding author.

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
