# Peer review of "Closed-Loop Uncertainty: The Evaluation and Calibration of Uncertainty for Human–Machine Teams under Data Drift"

_entropy, 2023, doi:10.3390/e25101443_

Round 1
Reviewer 1 Report
This manuscript proposes a stochastic framework by which an uncertainty model may be evaluated iteratively as a probability of machine correctness. In terms of topic, it is suitable to the Journal. In terms of methodological approach and presentation, I believe that the reported study is valuable in itself, but it suffers from certain disadvantages. Thus, the authors are advised to consider the probability to address the following remarks.
Remarks:
1. Please clarify the statement that “uncertainty can be used to manage cognitive load on the user by maintaining balance between exploration and exploitation” (l. 32-34).
2. Please clarify the following question: How many analysts were involved in this study? Were they naive with respect to the goal of the study? In one phase, is a “human-placed threshold” (denoted by the blue line in Fig. 2) determined based only on the opinion of one analyst or several of them?
3. In general, naive Bayes classification is based on the assumption of conditional independence, i.e, the assumption that likelihoods of features are independent for given class c. If this assumption does not hold for actual data used to train a Bayesian classifier, it may affect the performance of the system. Particularly related to this research, in the first paragraphs of Section 4.2, the authors define a set of features. Please discuss to what extent does the naive Bayes assumption hold in this case.
4. From the first paragraph of Section 4.3, it is not clear why a naive Bayes classifier has been taken as a baseline model. Please clarify.
5. Parameter T is defined in lines 213-214, related to the noise signal: \cos(T\pi). In lines 256-258, it is stated that variable T denotes the placement tolerance (which is defined in l. 258-260). Is it really the same parameter in both cases? Please explain.
6. The author state that “the posterior probability that a sample point in a trial should be given a label of 1 may be calculated”. However, as it is usually the case when a naive Bayes approach is applied (and keeping in mind that the authors apply a commonly used R package), I assume here that the authors do not consider a posterior probability, but rather a product of corresponding prior probability and likelihoods. This should be clarified. While this may appear as a formal remark, I would like to explain my concern here. If we keep in mind that (i) the likelihood values may be greater than one, and (ii) that the confidence value (as defined in this manuscript, please see the next remark) is defined as the average of “posterior” probabilities, the authors should explain how the values defined in Eq. (5) belong to set {1, 2, 3}.
7. The definition of confidence given in 265-267 is not precise. The authors first state that the confidence calculated “generated from this value” (i.e., a “posterior” probability that a sample point in a trial should be given a label of 1), which is most probably not correct, because immediately after they state that the confidence is calculated “by taking the average of all these probabilities “ (I assume: “posterior probabilities”) within the placement tolerance of the machine-placed threshold.
Please reformulate these statements to avoid confusion.
8. Please explain Eq. (7). (Why (1-C_t) in the second condition?)
9. Why the authors use the term “accuracy” in line 314?
10. Please discuss the generalizability of the proposed approach, which was illustrated for a relatively simple two-class task.
11. Please define abbreviation CLU, RL, etc.
Reviewer 2 Report
This article addresses the issue of uncertainty assessment and calibration for human-machine teams under data drift conditions. It proposes an iterative assessment of uncertainty models as a stochastic framework for machine correctness probability. It integrates the feedback of machine correctness into the baseline Naive Bayes uncertainty model using reinforcement learning algorithms. Finally, a large number of experiments are conducted to validate the proposed method against traditional baseline methods. The article's structure is clear, and the methodology is well-founded. However, there are some questions and suggestions regarding certain technical solutions and research methods.
Main issues to be fixed:
a) The article mentions in the Discussion that for more complex problems, a more complex feature space may be necessary, hence requiring a finer-grained state-action space and higher-level optimization. Therefore, when building the reinforcement learning algorithm model, the authors may consider using DQN to address the "curse of dimensionality" issue with action values, or employ reinforcement learning algorithms for continuous action spaces to address the issue of insufficient granularity in the action space.
b) I have some questions regarding the human-machine team proposed in the article. In Figure 3, the framework for closed-loop uncertainty shows that human judgment has a feedback effect on the reinforcement learning algorithm's decisions. However, in the introduction of the closed-loop uncertainty method, it seems to only mention human feedback in the evaluation methodology section and does not clearly demonstrate the role of human feedback judgment in the closed-loop uncertainty method. The description of the overall method architecture is not comprehensive enough.
c) The experimental validation section compares the errors and variances of the baseline algorithm and the CLU algorithm at different tolerances. The results analysis shows that the baseline method is superior to the proposed algorithm at certain tolerances. How should one choose which algorithm to use in practical problems? The results of the "machine placement accuracy" method are not included in Table 1. Additionally, I hope the author can explain the rationale behind using mean error with tolerances ranging from 0.02 to 0.2 to evaluate algorithm performance.
Round 2
Reviewer 1 Report
The authors have adequately addressed most of the remarks from my previous review report and I believe that the manuscript has been sufficiently improved to warrant publication.
Author Response
Thank you very much for your remarks, as they have improved the presentation of the research.
Reviewer 2 Report
For Comment a), I hope to see research on a more fine-grained state-action space in the near future, which can be mentioned in the future work section.
As for Comment b), human feedback is primarily reflected in the reward function and is used to correct incorrect machine placements. This feedback loop idea is the highlight of this article. I suggest providing a more detailed description of the proposed model's closed-loop structure in Section 4.3, incorporating Figure 3.
Lastly, regarding Comment c), I'm glad you have adopted my suggestions and made comprehensive revisions.
